# Circulating Tumour DNA Guided Adjuvant Chemotherapy Decision Making in Stage II Colon Cancer—A Clinical Vignette Study

**DOI:** 10.3390/cancers15215227

**Published:** 2023-10-31

**Authors:** Yat Hang To, Peter Gibbs, Jeanne Tie, Jonathan Loree, Tamara Glyn, Koen Degeling

**Affiliations:** 1Personalised Oncology Division, Walter and Eliza Hall Institute of Medical Research, Melbourne, VIC 3000, Australia; 2Department of Medical Oncology, Peter MacCallum Cancer Centre, Melbourne, VIC 3000, Australia; 3Department of Medical Oncology, Western Health, St Albans, VIC 3091, Australia; 4Faculty of Medicine & Health Sciences, University of Melbourne, Melbourne, VIC 3000, Australia; 5Division of Medical Oncology, BC Cancer, Vancouver, BC V5Z 4E6, Canada; 6Department of Surgery, University of Otago, Christchurch 8011, New Zealand; tamara.glyn@cdhb.health.nz; 7Department of Surgery, Te Whatu Ora Health New Zealand Waitaha Canterbury, Christchurch 8140, New Zealand; 8Cancer Health Services Research, Centre for Cancer Research, Faculty of Medicine, Dentistry and Health Sciences, University of Melbourne, Melbourne, VIC 3000, Australia; 9Cancer Health Services Research, Centre for Health Policy, Melbourne School of Population and Global Health, Faculty of Medicine, Dentistry and Health Sciences, University of Melbourne, Melbourne, VIC 3000, Australia

**Keywords:** colon cancer, circulating tumour DNA, biomarker, chemotherapy, decision making

## Abstract

**Simple Summary:**

Circulating tumour DNA is a biomarker of significant research interest with a number of randomised controlled trials comparing a ctDNA-informed approach to adjuvant decision making in stage II colon cancer compared to standard of care. However, it is unknown if medical oncologist would recommend ctDNA testing in real-world clinical practice and how results may influence treatment recommendations. We presented medical oncologists with a series of stage II colon cancer clinical vignettes, demonstrating that surveyed participants are willing to organise ctDNA testing in most clinical scenarios. Importantly, oncologists were more likely to recommend adjuvant chemotherapy and escalate treatment following a positive result (i.e., detectable ctDNA). Following a negative result, oncologists were inclined to de-escalate or avoid chemotherapy. The results demonstrate that ctDNA testing can influence treatment decision making and can also be utilised in future economic evaluations to help secure access to testing.

**Abstract:**

Circulating tumour DNA (ctDNA) is a promising biomarker that may better identify stage II colon cancer (CC) patients who will benefit from adjuvant chemotherapy (AC) compared to standard clinicopathological parameters. The DYNAMIC study demonstrated that ctDNA-informed treatment decreased AC utilisation without compromising recurrence free survival, but medical oncologists’ willingness to utilise ctDNA results to inform AC decision is unknown. Medical oncologists from Australia, Canada and New Zealand were presented with clinical vignettes for stage II CC comprised of two variables with three levels each (age: ≤50, 52–69, ≥70 years; and clinicopathological risk of recurrence: low, intermediate, high) and were queried about ctDNA testing and treatment recommendations based on results. Sixty-four colorectal oncologists completed at least one vignette (all vignettes, *n* = 59). The majority of oncologist were Australian (70%; Canada: *n* = 13; New Zealand: *n* = 6) and had over 10 years of clinical experience (*n* = 41; 64%). The proportion of oncologists requesting ctDNA testing exceeded 80% for all vignettes, except for age ≥ 70 and low-risk disease (63%). Following a positive ctDNA result, the proportion of oncologists recommending AC (*p* < 0.01) and recommending oxaliplatin-based doublet (*p* < 0.01) increased in all vignettes. Following a negative result, the proportion recommending AC decreased in all intermediate and high-risk vignettes (*p* < 0.01).

## 1. Introduction

The benefits of adjuvant chemotherapy (AC) following the resection of stage II colon cancers (CC) remains unclear, as trials and meta-analyses have failed to identify a subset of patients that consistently derive a survival benefit [1,2]. Current consensus guidelines recommend that AC may be offered to patients with proficient mismatch repair (pMMR) and high-risk clinicopathological parameters, including T4 staging, the presence of lymphovascular (LVI) or perineural invasion (PNI), poor tumour differentiation, inadequate lymph node (LN) examination, or tumour obstruction or perforation [3,4]. However, there is no compelling evidence that any of these factors predict a benefit from AC.

One strategy to refine patient selection is the use of molecular and genomic biomarkers to better define recurrence risk compared to standard clinicopathological parameters. Circulating tumour DNA (ctDNA) are tumour-derived genetic fragments released into peripheral circulation that can be detected in the plasma of patients with cancer. Tie et al. (2016) were the first to describe the prognostic value of ctDNA as a blood-based biomarker in an observational study of patients with stage II CC [5]. In a cohort of patients not treated with AC, 79% of those (*n* = 11/14) with detectable ctDNA (i.e., ctDNA-positive) following curative intent surgery had recurred compared to 7% of patients with negative ctDNA (hazard ratio (HR) 18, *p* < 0.001) [5]. Subsequent cohort studies by other investigators consistently demonstrated that patients with post-operative detectable ctDNA are at markedly increased risk of recurrence [6,7,8,9]. However, most studies were retrospective in nature or had blinded ctDNA results, so the influence of testing on AC prescription is unknown. These early findings informed the design of the DYNAMIC study in stage II CC patients, representing the world’s first randomised controlled study directly comparing a ctDNA-informed approach to a standard of care (SOC) for AC selection in stage II colon cancer patients [10]. In the ctDNA arm, only positive patients were offered AC, but the specific regimen was at the treating physician’s discretion. The study, first published in June 2022, reported that the ctDNA-guided approach resulted in fewer patients receiving AC (15 vs. 28%; relative risk 1.82, 95% confidence interval (CI) 1.25 to 2.65) without compromising 2-year recurrence-free survival (RFS) [10].

Internationally, randomised trials of ctDNA-informed AC, such as CIRCULATE-PRODIGE (EudraCT 00002019-000935-15), CIRCULATE-AIO (NCT 04089631) and COBRA (NCT 04068103), are also recruiting stage II CC patients [11,12,13]. Positive results could further support the use of ctDNA in clinical care. However, as AC prescription in these studies are constrained by protocol, participating clinicians often have limited discretion in assigning treatment. For example, most studies mandate all ctDNA-negative patients be spared AC. Therefore, the clinical scenarios in which oncologists would order ctDNA testing and how these results would shape treatment recommendation in routine clinical practice remain unknown. The uptake of testing would have important implications for healthcare resource allocation and economic evaluations. We conducted an online survey of oncologists actively treating patients with stage II CC, focusing on how they would utilise ctDNA results to guide AC decision making in clinical practice.

## 2. Methods

Medical oncologists from Australia, Canada and New Zealand were invited via electronic mail to complete an online survey. Oncologists were identified through their membership in collaborative groups, including the Medical Oncology Group of Australia (MOGA), the Canadian Cancer Trials Group (CCTG) and the New Zealand Society for Oncology (NZSO). The survey was first circulated to the Australian participants in November 2022 and subsequently to Canadian and New Zealand participants in February and March 2023, respectively.

The survey was divided into three sections (see Appendix A). Part 1 elicited the participant demographics, including the number of years they had been practicing as a qualified oncologist and an estimate of the number of newly diagnosed patients with stage II CC treated per year. Part 2 investigated the participants’ familiarity with the concept of ctDNA and the results of the DYNAMIC study. Participants were also asked to provide a maximum cost that patients should pay for ctDNA testing privately if the cost was not covered by government-funded reimbursement schemes.

Part 3 contained a series of nine clinical vignettes representing a patient being considered for AC following the resection of their stage II CC. Each vignette was constructed of two variables with three levels each, and all permutations were presented to participants (see Table 1). Age was included as a variable previous real-world studies have indicated that older patients are less likely to receive AC, suggesting that oncologists are less likely to recommend AC in certain age groups. The levels for recurrence risk were adapted from the European Society Medical Oncology (ESMO) clinical practice guidelines for localised colon cancer [4]. All patients were considered to have no other medical co-morbidities and had adequate functional status. Oncologists were queried if they would order ctDNA testing (if available at minimum cost to patients) and for their AC recommendation in three scenarios: (1) vignette with age and clinicopathological information alone (pre-ctDNA testing), (2) ctDNA undetected (ctDNA-negative), and (3) ctDNA detected (ctDNA-positive). The specific recommended regimen (5-fluorouracil (5-FU), capecitabine, 5-FU and oxaliplatin (FOLFOX), or capecitabine and oxaliplatin (CAPOX)) and duration (3 or 6 months) were also recorded. Oncologists were considered to have escalated treatment following ctDNA testing if they modified their recommendation from no AC to AC, added oxaliplatin to fluoropyrimidine (i.e., single to doublet) or prolonged treatment duration. Conversely, de-escalation was defined as switching from any AC to no AC, dropping oxaliplatin (i.e., doublet to single), or shortening the duration.

All responses were anonymous. Chi-square testing was used to describe differences in proportions, and a *p*-value < 0.05 was considered statistically significant. To ensure complete of reporting, responses for the Checklist for Reporting Results of Internet E-Surveys (CHERRIES) is provided in the Appendix A [14].

## 3. Results

The survey was circulated to 721 medical oncologists with 64 (response rate: 8.8%) completing at least 1 vignette and 59 (92% of respondents) completing all vignettes. However, the distribution amongst Australian oncologists (*n* = 396) was not specific to colorectal cancer specialists. Full demographic characteristics are available in the Appendix A. The majority of oncologist were Australians (*n* = 45, 70%; Canada: *n* = 13, 20%; New Zealand: *n* = 6, 10%), had over 10 years clinical experience (*n* = 41; 64%) and saw ≤ 10 new stage II patients per year (*n* = 39, 61%). Regarding the place of practice, 20.3% (*n* = 13) held their major role in a private practice, 28.1% (*n* = 18) in rural/regional practices and 30.7% (*n* = 19) in specialist oncologist centres.

All oncologists stated that they were familiar with the concept of ctDNA, with 96.9% aware of the DYNAMIC study publication and 85.9% having read the publication. When presented with the statement “Patients with detectable ctDNA following resection of stage II colon cancer are at higher risk of recurrence compared to patients with undetectable ctDNA”, 71.9% (*n* = 46) of oncologists strongly agreed, 23.5% (*n* = 15) agreed, 3.1% (*n* = 2) were uncertain, and 1.5% (*n* = 1) disagreed. The median maximum cost that oncologists believed patients should pay privately for a ctDNA test was AUD500 (range: 0 to 5000)

### 3.1. Adjuvant Chemotherapy Recommendation—Pre-ctDNA Testing

Table 2 demonstrates the AC recommendation for each vignette based on age and the clinic-pathological assessment of risk alone, reflecting current clinical practice. The proportion of oncologists recommending AC increased with clinicopathological risk levels (low vs. intermediate = *p* < 0.01; low vs. high = *p* < 0.01; intermediate vs. high = *p* < 0.01) and decreased when comparing elderly (≥70 years) to young (≤50 years) patients (*p* = 0.02). The oldest age group was less likely to be offered AC (88.7% vs. 100%) or doublet chemotherapy (21% vs. 56.6%) compared to the youngest age group, even in the high-risk population (all *p* < 0.01).

For each vignette, years in clinical practice, the number of newly diagnosed stage II CC seen by year and types of practice did not predict AC recommendation (see Appendix A.

The proportion of oncologists that would request ctDNA testing exceeded 80% for all clinical scenarios aside from the vignette that represented a patient aged ≥ 70 and with a low risk based on standard clinic-pathological factors. Notably, this is the scenario for which no oncologist would recommend AC pre-ctDNA testing, indicating a reluctance to order testing when the effectiveness of AC is perceived to be limited. The highest proportion of testing requests were observed in the intermediate risk patients (>90%). This patient group also had the most diverse range of pre-ctDNA treatment recommendations. This highlights a patient group that should be specifically considered in future trials as the results indicate a need for additional risk stratification methods beyond standard clinic-pathological assessment.

### 3.2. Adjuvant Chemotherapy Recommendation—Post-ctDNA Testing

Figure 1 demonstrates the proportion of oncologists recommending AC based on ctDNA results. Following a negative result, the proportion of oncologist recommending AC significantly decreased in all intermediate- (*p* < 0.01) and high-risk scenarios (*p* < 0.01). This demonstrates a willingness to avoid treatment even in patients that are traditionally thought to derive the most clinical benefit from AC. Years in clinical practice, the number of newly diagnosed stage II CC seen by year, and types of practice did not predict AC recommendation (Appendix A). 

Following a positive ctDNA result, the proportion of oncologists recommending AC significantly increased in all scenarios (<0.01) aside from high-risk patients aged below 70 years, noting that ≥ 90% of oncologist had already recommended AC prior to ctDNA testing. This suggests that surveyed oncologists hold a strong belief that a positive ctDNA result denotes a high risk of recurrence and a willingness to prioritise ctDNA over traditional clinicopathological parameters. The interaction between ctDNA results and clinico-pathological risk factors needs to be elicited in future studies to determine the optimal roles they may play in guiding AC recommendation. Additionally, years in clinical practice, the number of newly diagnosed stage II CC seen by year, and types of practice did not predict AC recommendation (Appendix A).

Figure 2 demonstrates the AC regimen recommended following negative and positive ctDNA results. Tabulated results are available in the Appendix A. Following ctDNA detection, the proportion of oncologists recommending doublet chemotherapy increased (*p* < 0.01 for all vignettes), whereas this generally decreased following a negative ctDNA result (*p* < 0.01 for all intermediate- and high-risk vignettes). The impact of doublet AC in ctDNA-positive patients needs to be elicited in clinical trials, as does its role for further escalation, such as the incorporation of irinotecan. Table 3 demonstrates the proportion of oncologists whose AC recommendations were de-escalated or escalated following ctDNA testing. AC treatment recommendations significantly changed depending on ctDNA results (*p* < 0.01) for all vignettes, demonstrating the oncologists’ willingness to adopt results into their recommendations. The proportion of oncologist recommending specific regimens and durations are available in the Appendix A

## 4. Discussion

To our knowledge, this survey represents the first analysis of medical oncologists’ attitudes to incorporating ctDNA testing into their AC decision making in stage II colon cancer following the publication of the DYNAMIC study. The results demonstrate a variable approach to AC in routine care and that a high proportion of clinicians are willing to order ctDNA testing in almost all clinical situations. ctDNA results significantly impacted treatment recommendations, with oncologists more likely to recommend AC and escalate to doublet regimens following a positive result. Conversely, following a negative result, clinicians were more likely to not recommend AC. The study findings reflect real-world attitudes to ctDNA-testing and the influence on treatment recommendations. It is important to note that these results are not meant to serve as definitive recommendations or support the adoption of ctDNA into routine clinical practice. However, considering the strong interest in testing, it underscores the importance of further studies in determining the efficacy of ctDNA testing in personalising AC treatment, particularly in evaluating the potential to inform chemotherapy avoidance or intensification.

The pre-ctDNA recommendations reflect the ongoing uncertainty in regards to the role of AC in stage II CC. Respondents were predictably more likely to recommend AC with increasing linic-pathological risk levels, validating this aspect of the vignette design. The greatest variation in recommendation was observed in the intermediate-risk group, reflecting a significant variation in individual clinician interpretation of the data on recurrence risk outside of tumour staging and inadequate lymph node sampling. When AC was recommended, clinicians favoured single-agent regimens in all vignettes, reflecting the ongoing uncertain benefit of adding oxaliplatin, even in high-risk stage II CC [15]. Oncologists were also less likely to recommend AC in older patients (≥70 years) despite evidence that elderly patients derived a similar benefit in terms of improved survival and time to recurrence without a significant increase in toxicity with single-agent fluorpyrmidine [16]. Our data are consistent with those obtained in previous analyses on AC use in stage II colon cancer in which there was a similar reluctance to subject older patients to AC [17,18].

In a previous survey of American oncologists by Savill et al. (2022), pre-dating the DYNAMIC study publication, 25% of the 55 respondents indicated that they were currently using ctDNA to guide post-resection AC decisions, although this was not limited exclusively to patients with stage II CC [19]. These results differ from our findings, with a far higher proportion of oncologists indicating they would order ctDNA testing in most clinical situations. It is possible that this disparity is partly due to our survey explicitly stating that ctDNA testing was available at minimal cost to patients. In our survey, the only scenario in which fewer than 80% of respondents would consider testing was in older, clinicopathological low-risk patients, likely reflecting uncertain benefit and a perceived higher risk of toxicities in the elderly population. However, the majority of oncologists recommended AC when post-operative ctDNA was detected, even in patients determined to be at very low risk of recurrence based on traditional risk factors. This highlights a willingness to accept ctDNA as a more accurate measure of recurrence risk than current clinico-pathological risk factors derived from previous observational data [5].

Our results also indicate that in ctDNA-positive patients, oncologists were more likely to recommend adding oxaliplatin to fluoropyrimidine, a strategy supported by the DYNAMIC study, demonstrating that amongst ctDNA-positive patients, the receipt of doublet AC achieved a 3-year RFS of 93%, contrasting with the very high recurrence rates observed in earlier observational studies [10]. Some respondents also prolonged AC duration, presumably extrapolating findings from the IDEA collaboration, which indicated a benefit of six versus three months of treatment in clinico-pathologically high-risk patients [20].

Our surveyed oncologists were also willing to de-escalate treatment in the setting of undetectable ctDNA, including not recommending AC, even in high-risk patients. This recommendation is supported by findings from the DYNAMIC study, where a low risk of recurrence (7%) was reported in untreated ctDNA-negative patients, with study authors concluding that AC can be safely avoided in this cohort [10]. In post hoc analyses, 3-year RFS appeared to be increased in untreated ctDNA-negative patients with no high-risk clinico-pathological features compared to those with at least one feature, mainly driven by T4 disease (96.7% vs. 85.1%; HR 3.04, 95% CI 1.26 to 7.34) [10]. As the design of the DYNAMIC trial did not allow for the treatment of ctDNA-negative patients, the benefits of AC in the subset of ctDNA-negative patients with traditional high-risk features are unknown. This uncertainty is reflected in our results, in which a proportion of clinicians still recommended AC in ctDNA-negative, high-risk patients, but the utilisation of doublet AC decreased, suggesting at least a willingness to de-escalate treatment intensity.

Savill et al. in their survey also identified the key barriers to the more widespread uptake of ctDNA testing, including reimbursement issues (56% of respondents), insufficient clinical evidence (46%) and limited familiarity (28%) [19]. Our respondents were all familiar with the concept of the ctDNA, with the vast majority being aware of the DYNAMIC study and having read the associated publication. The DYNAMIC study also provides gold-standard randomised controlled trial clinical evidence to support the use of ctDNA to inform the treatment of stage II CC. Many other studies, such as CIRCULATE-PRODIGE, CIRCULATE-AIO and COBRA, are recruiting patients with stage II CC, and these results could further strengthen the case for the routine use of ctDNA in this patient population [11,13,21].

Regarding reimbursement, the countries included in our survey each have public health care systems through which affordable access to new health technologies, such as ctDNA, is traditionally provided via government-run and taxpayer-funded schemes. Although clinically effective health technologies should be made available to all patients, equitable access in many countries is dependent on securing reimbursement, which in turn requires the demonstration of cost-effectiveness through health economic evaluations. An early cost-utility analysis based on the early cohort study of stage II CC patients by Tie et al. (2021) demonstrated the potential of ctDNA-guided AC decision making to be cost-effective compared to SOC but the authors acknowledged uncertainties surrounding the evaluation due to the early nature of the clinical evidence [22]. Further economic evaluations based on the DYNAMIC trial and other studies will help build on this early evidence. The results of our vignette survey could help increase the robustness of such evaluations by providing estimates of potential real-world uptake and how the results might influence AC recommendations in the context of current practice. Demonstrated cost-effectiveness may assist in securing reimbursement and ensuring access, but further demonstrated efficacy of ctDNA testing in clinical trials is the first critical step.

This study has certain limitations. Firstly, the number of respondents is modest, limiting the generalisation of the results to a broader landscape of oncological practice. Additionally, it is crucial to recognise the potential presence of selection bias, as the inclusion of participants from oncological societies and research interest groups may suggest a cohort who is more inclined towards the earlier adoption and utilisation of newer health technologies. However, when considering pre-ctDNA testing recommendations, there was no significant difference in response when stratifying by country, years in practice, and if oncologists practised in private or regional care. Following ctDNA results, there was again no difference in recommendations for AC when considering these stratifications. Most respondents were experienced clinicians (majority having ≥10 years of clinical experience), and most worked in general hospital settings, suggesting the responses may be reflective of general oncological practice. Secondly, the vignettes can be considered an oversimplification of real-world AC decision making. Although we adapted the ESMO guidelines to formulate our clinicopathological risk levels, other guidelines, such as the National Comprehensive Cancer Network (NCCN) guidelines, do not stratify the clinico-pathological risks into similar levels, rather simply stating that these risk factors need to be considered [3]. The vignettes also do not capture other real-world considerations, such as patient attitudes, co-morbidities, and more nuanced approaches to assessing toxicity risk beyond chronological age alone, such as formal geriatric evaluations. Testing for genetic variations in the dihydropyrimidine dehydrogenase gene (DPYD) can also inform treatment risk [23,24]. However, incorporating more variables and levels would have exponentially increased the number of vignettes presented to oncologist and decreased the likelihood of completed responses. Given the consistency of our results, it is also debatable if other variables would have significantly influenced the recommendations. Ultimately, if reimbursement for ctDNA testing in this patient population is secured and real-world data on its use in practice becomes available, further research into real-world testing patterns and decision making is warranted. Further research may also be performed into patient attitudes towards ctDNA testing to assess their understanding of this biomarker and how they would incorporate the testing results into their decision making.

## 5. Conclusions

This clinical vignette study demonstrated that medical oncologists are willing to order ctDNA testing to guide adjuvant chemotherapy decision making for patients with stage II colon cancer. Importantly, the surveyed oncologists indicated that ctDNA results would influence treatment recommendations in most patients, with treatment escalation for patients with detectable ctDNA, and the de-escalation or avoidance of adjuvant chemotherapy in patients with undetectable ctDNA.

## Figures and Tables

**Figure 1 cancers-15-05227-f001:**
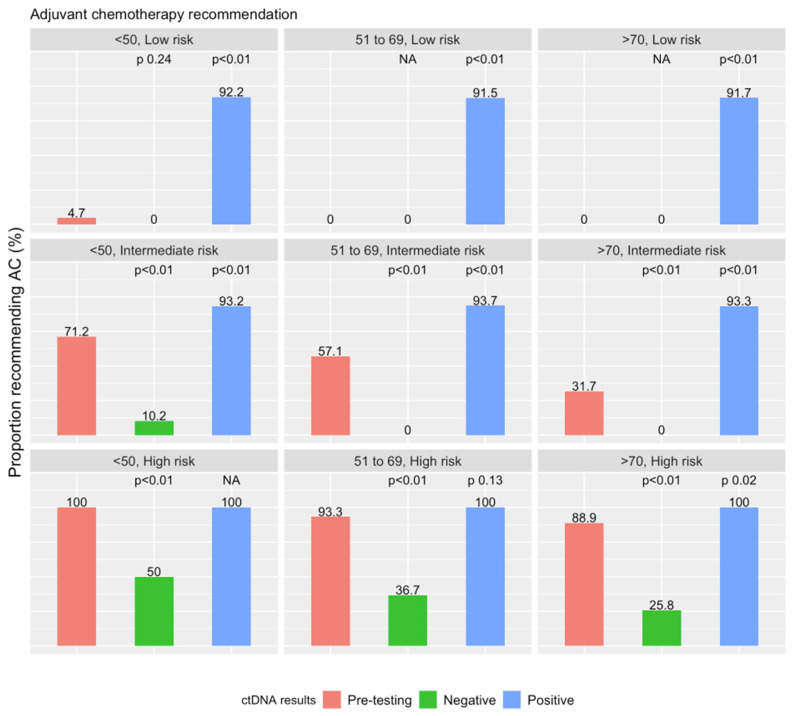
Comparing recommendation for adjuvant chemotherapy following ctDNA testing. Results of the Chi-square testing comparing the proportion of oncologists recommending chemotherapy pre- and post-ctDNA testing is also demonstrated.

**Figure 2 cancers-15-05227-f002:**
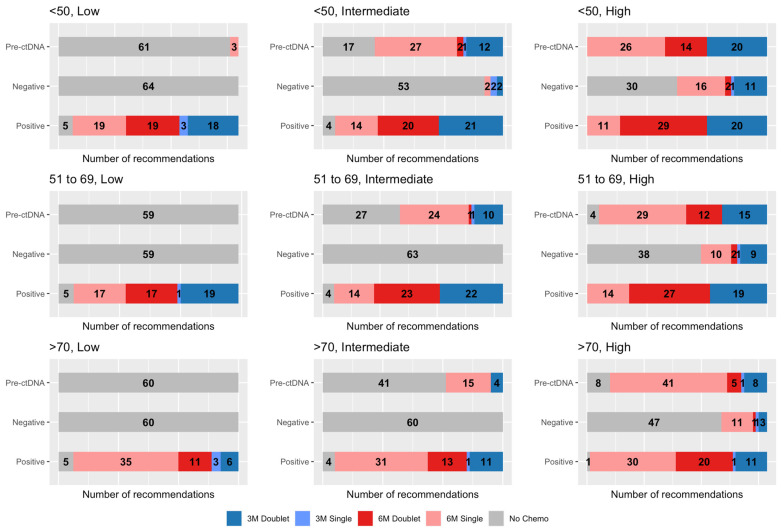
Adjuvant chemotherapy regimen recommendations pre-ctDNA testing compared to negative and positive results for each clinical scenario.

**Table 1 cancers-15-05227-t001:** Clinical vignette structure, constructed of two variables with three levels each, resulting in nine possible vignettes in total, each made up of an age variable and clinico-pathological variable. The levels for clinico-pathological risk for recurrence were adapted from the European Society Medical Oncology (ESMO) clinical practice guidelines for localised colon cancer.

	Variable 1: Age	Variable 2: Clinico-Pathological Risk for Recurrence
Level 1	≤50 years	High-risk—defined as the presence of any of the following features: Less than 12 lymph nodes examined pT4 stage including perforation Multiple (>1) intermediate risk features
Level 2	51 to 69 years	Intermediate-risk—defined as mismatch repair proficient (pMMR) tumour and the presence of only one of the following features: Poor/high-grade tumour differentiation Vascular invasion Lymphatic invasion Perineural invasion Tumour presentation with obstruction
Level 3	≥70 years	Low-risk—defined as one of the following: pMMR or MMR-deficient (dMMR) with no, high- or intermediate-risk features dMMR tumour with only ONE intermediate risk feature

**Table 2 cancers-15-05227-t002:** Adjuvant chemotherapy recommendation for each vignette based on age and clinic-pathological assessment of risk. Single agent regimen includes 5-fluoruoracil or capecitabine alone. Doublet regimen includes FOLFOX (5-flouoruacil and oxaliplatin) or CAPOX (capecitabine and oxaliplatin).

Clinicopathological Risk	Low			Intermediate			High		
Age	≤50	51 to 69	≥70	≤50	51 to 69	≥70	≤50	51 to 69	≥70
No. of responses	64	59	60	59	63	60	60	60	62
Recommend AC, *n* (%)	3 (4.7)	0 (0)	0 (0)	42 (71.2)	36 (57.1)	19 (31.7)	60 (100)	56 (93.3)	55 (88.7)
Specific regimen recommended, *n* (%)									
3 M Single	0 (0)	0 (0)	0 (0)	1 (1.7)	1 (1.6)	0 (0)	0 (0)	0 (0)	1 (1.6)
6 M Single	3 (4.7)	0 (0)	0 (0)	27 (45.8)	24 (38.1)	15 (25)	26 (43.3)	29 (48.3)	41 (66.1)
3 M Doublet	0 (0)	0 (0)	0 (0)	12 (20.3)	10 (15.9)	4 (6.7)	20 (33.3)	15 (25)	8 (12.9)
6 M Doublet	0 (0)	0 (0)	0 (0)	2 (3.4)	1 (1.6)	0 (0)	14 (23.3)	12 (20)	5 (8.1)
No chemotherapy	61 (95.3)	59 (100)	60 (100)	17 (28.8)	27 (42.9)	41 (68.3)	0 (0)	4 (6.7)	7 (11.3)
Order ctDNA testing, *n* (%)	54 (84.4)	49 (83.1)	38 (63.3)	54 (91.5)	60 (95.2)	54 (90)	49 (81.7)	49 (81.7)	55 (88.7)

Abbreviations: 3 M = 3 months, 6 M = 6 months, ctDNA = circulating tumour DNA.

**Table 3 cancers-15-05227-t003:** Changes to adjuvant chemotherapy recommendation following negative and positive ctDNA results. The *p*-value represents the results of Chi-square testing to compare changes in treatment recommendations (no change vs. de-escalation/escalation) and ctDNA results.

	Clinicopathological Risk	Low			Intermediate			High		
	Age	≤50	51 to 69	≥70	≤50	51 to 69	≥70	≤50	51 to 69	≥70
	No. of responses	64	59	60	59	63	60	60	60	63
Negative ctDNA	No change, *n* (%)	61 (95)	59 (100)	60 (100)	21 (35)	27 (43)	41 (68)	22 (37)	20 (33)	22 (35)
	De-escalate, *n* (%)	3 (5)	0	0	38 (65)	36 (57)	19 (32)	38 (63%	40 (67)	41 (65)
	Escalate, *n* (%)	0	0	0	0	0	0	0	0	0
Positive ctDNA	No change, *n* (%)	5 (8)	5 (8)	5 (8)	22 (59)	15 (24)	13 (22)	37 (62)	33 (55)	38 (60)
	De-escalate, *n* (%)	0	0	0	0	0	0	0	0	0
	Escalate, *n* (%)	59 (92)	54 (92)	55 (92)	37 (41)	48 (76)	47 (78)	23 (38)	27 (45)	25 (40)
*p*-value		<0.01	<0.01	<0.01	<0.01	<0.01	<0.01	<0.01	<0.01	<0.01

## Data Availability

The data generated during and/or analysed during the current study are available from the corresponding author, Y.H.T., upon reasonable request.

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
