# Peer review of "Circulating Tumour DNA Guided Adjuvant Chemotherapy Decision Making in Stage II Colon Cancer—A Clinical Vignette Study"

_cancers, 2023, doi:10.3390/cancers15215227_

Round 1
Reviewer 1 Report
Comments and Suggestions for Authors
The manuscript is well write and the concept are explain clearly.
Reviewer 2 Report
Comments and Suggestions for Authors
The vignette study conducted by To et al on the topic of medical oncologists' willingness to use circulating tumor DNA (ctDNA) testing in real-world clinical practice and how the results may influence treatment recommendations in stage II colon cancer is sound and concise. Although I believe this study makes a significant contribution to the clinical field, I have some comments for authors:
Comments:
1. The introduction could provide more information on the limitations of existing literature on ctDNA testing and the rationale for this study.
2. The results section could discuss the implications of the findings in more detail, such as the potential impact on clinical practice and future research.
3. The discussion section could discuss the limitations of the study in more detail, such as the small sample size and the potential for selection bias.
Comments on the Quality of English LanguageMinor editing and careful proofreading required.
Reviewer 3 Report
Comments and Suggestions for Authors
The manuscript is interesting and original. The authors faced on the oncologist attitude to offer adjuvant chemotherapy in stage II colorectal cancer patients. The design of the study is clear and the results well written. However, I suggest some minor changes with the aim to improve the value of the manuscript.
Why the authors have decided to stratify the patients according to age? There is low evidence that age is a risk factors and mandatory in AC decision. This choiche makes the results a little bit confusing. For the same reason Figures 2 an 3 are very difficult to read and understand. It is possible to simplify it?
The authors better should better underline that describing a real-life attitude don’t means that cDNA technology should be systematically used into the clinical practice. More evidence is needed on this topic and caution should be used on this. Moreover, a technology to be proposed should be available in the health care systems, outside the costs. Please provide some statements and comments on these two topics.
